# OpenReview forum: "PlatoLM: Teaching LLMs  via a Socratic  Questioning User Simulator"
_ICLR.cc/2024/Conference — ICLR 2024 Conference Withdrawn Submission_

### Official Review · Reviewer_GXfd · 2023-10-31

**Soundness:** 2 fair
**Presentation:** 3 good
**Contribution:** 1 poor
**Rating:** 6
**Confidence:** 5

**Summary:**

The paper proposes a new method to train large language models (LLMs) using a trainable user simulator called "Socratic" to synthesize high-quality conversational data.

- Authors train a "Socratic" model on human questions from ShareGPT to mimic the questioning skills of real users. Socratic Can have free conversations or conversations which start from seeded example questions.
- Authors generate dataset "SocraticChat" via conversations between Socratic and ChatGPT.
- Authors fine-tuned PlatoLM (like Plato) on SocraticChat as the system agent.
- Authors performed some analysis on the quality of the fine-tuned PlatoLM and the generated dataset SocraticChat.

**Strengths:**

The paper has a clean presentation, and it's easy to follow the authors' ideas.

**Weaknesses:**

- **Problem Motivation:** The goal of the current work is to make it easier and cheaper to produce synthetic dialogues for fine-tuning language models for chat applications. The specific approach here is to replace the human side with an LM trained on human queries. This trained LM then interacts with ChatGPT to bootstrap more data. While the approach can produce interesting artifacts such as the dialogue dataset, I don't think the research goal -- making bootstrapping data from ChatGPT easier -- is a scientific problem.
- **Performance**: After reading the paper, it's unclear to me whether there are real gains from first training a human query simulator and then using its simulated data. For instance, in Table 2 with the same number of examples, PlatoLM-7b doesn't outperform Vicuna-7B with 10k examples for MT-bench.
- **Understanding**: After reading the analysis section (Sec 5), it's still unclear to me that if there's a benefit from using Socratic simulated data, what would be an intuitive reason for that? I can imagine the technique useful when the number of total human queries is small, where fine-tuning on human queries helps the query model learn the style. The pretrained base has a lot of knowledge; thus the fine-tuned query model can produce much more diverse query content. But small human data setting is not explored.

I read the author response and have updated my score.

**Questions:**

For vicuna-7b, authors should clarify the exact version, i.e., is it v1.3 or v1.1, or something else.

---

> ### Author Response · Authors · 2023-11-17
>
> ## Motivation
>
> We are very sorry for the misunderstanding. Our motivation focuses on how we can **simulate** human questioning better and how the human questioning pattern **benefits** the response quality after witnessing the success of Vicuna, absolutely not on how to bootstrap data from ChatGPT more easily. In fact, ChatGPT can be replaced by any other stronger response models. We used it just because of the high-quality answers. The reason why we conducted the scaling experiments is also to explore whether the benefits of human questioning pattern to response model can be scalable.
>
> In other words, the dataset is only a **by-product** of our paradigm.
>
> ---
>
> ## Performance
>
> It may be caused by the confusing layout of our tables, please review the revised Tables 1 and 2 again.
>
> **If we use randomly sampled 10K conversations**
>
> Compared to the static simulation (Baize、UltraLM), our model outperforms it in alpaca-eval、vicuna-bench、mt-bench.
>
> Compared to Vicuna, Our model outperforms it in alpaca-eval, vicuna-bench, and mt-bench's pair-wise evaluations. In fact, we just **lag behind Vicuna in the single evaluation** judged by machine and we have explained it in 4.2.2 Automatic Evaluation.  Specifically, the author of the MT-Bench paper gave us the example: if one model is really bad in the math category, it'll **get penalized more by the low scores in single answer grading than in pairwise win rate setup**.  Our model performs badly **in the math and extraction category** (in Appendix C. Analysis on Domain).
>
> In fact, using randomly sampled 10K conversations is a **preliminary experiment**. In the beginning, we envisaged Vicuna as our ceiling.  After conducting **the scaling experiments** based on LLaMA1-LLaMA1（in Sec.5.3）, we find that the teaching ability of our paradigm can be **scalable** with the data volumes. Using a 100K sample (according to **Reviewer FWRq and Z8HQ**) can beat the Vicuna-10K in MT-Bench's **single evaluation**（5.85 v.s. .5.9598±0.0339）.
>
> **If we use 50k samples based on LLaMA1-LLaMA2 (in Table2)**
>
> Under the same backbone LLaMA2, PlatoLM beats Vicuna-7b-v1.5 and LLaMA-2-7b-chat in MT-Bench and Alpaca-Eval. The former was trained on 125K samples with a context length of 16K, the latter was trained on 1100K samples using SFT and RL.
>
> After conducting the pairing experiments in Appendix E according to **Reviewer Z8HQ**, we find **the different pairing backbone of Socratic and PlatoLM** will be more adaptive to our paradigm（see table below）. With only 28.5K samples, Plato-7B surpassed Vicuna-13B-v1.5 and LLaMA-2-13B-chat in MT-Bench.
>
> | Q/A     | LLaMA-1       | LLaMA-2       | Mistral       |
> | ------- | ------------- | ------------- | ------------- |
> | LLaMA-1 | 5.7506±0.0314 | 6.0920±0.0515 | 6.4187±0.0470 |
> | LLaMA-2 | 5.8775±0.0205 | 5.9900±0.0163 | 6.6838±0.0464 |
> | Mistral | 5.9094±0.0261 | 6.1744±0.0199 | 6.3250±0.0406 |
>
> ---
>
> ## Understanding
>
> **Rationale of PlatoLM**
> In our paper, the intuitive reason is that **the human-like questioning pattern** can elicit the high-quality responses better in real-time dialogs. In the human-to-machine interaction, humans dominate the conversation. Hence, a **more human-like questioning pattern** will bring a **richer topic structure**, such as topic drifting and shifting, leading to higher topic diversity. This was all quantified in revised Sec.5.4.
>
> Compared to static stimulation, the questions posed by Socratic are more human-like, and the topic diversity is higher. （in Sec 5.2 Table 4）.
>
> Compared to real human-to-machine dialogs, **the different pairing backbone and the scalability of data** can make our paradigm better.
>
>
> **For the small human data setting**
> The newly-added scaling experiments (with 100k examples) demonstrated that the performance of the response model does not saturate in MT-bench (especially in the second turn) as the data volume increases to **100K**.
>
> Using fewer training examples might negatively affect the user simulator as it starts from a *base* model; the *base* might be originally poor in questioning, and it needs some data for training, so we used **20K samples** to fine-tune the simulator.  Although we also used ShareGPT-gpt4-6K to train the simulator and found it works well when inferencing, we're not clear on your definition of the small human data setting. In other words, is 6K small enough? If 6K is small enough, we will control the questions without repetition and test whether the questioning model can ask more questions than 6K.
>
> ---
>
> ## Version of vicuna-7b
> The baseline version is V1.5, see https://huggingface.co/spaces/lmsys/chatbot-arena-leaderboard. Vicuna gets a score of 6.17 and is linked to  https://huggingface.co/lmsys/vicuna-7b-v1.5 . we have added the version and the training type in the revised version.
>
> p.s. All sections with ± symbols on our supplementary experimental tables indicate that the results were rated 5 times by gpt4 and the standard deviation was calculated.

---

> ### Author Response · Authors · 2023-11-22
> **Polite inquiry**
>
> We sincerely apologize for the misunderstanding caused by our mistake.
> We kindly request you to reconsider the scores once we have addressed your concerns and conducted additional experiments. Alternatively, if there are any remaining concerns that we have not addressed, please let us know. We would be more than happy to provide any necessary materials and experimental results, or clarify any aspects that may have caused confusion.
>
> Thank you for your time and consideration!

---

### Official Review · Reviewer_FWRq · 2023-11-01

**Soundness:** 3 good
**Presentation:** 3 good
**Contribution:** 3 good
**Rating:** 6
**Confidence:** 3

**Summary:**

The paper introduces a novel approach where they train a user simulator called 'Socratic' using real user data. They use 'Socratic' to generate synthetic user-system conversation data when interacting with ChatGPT. The resulting dataset is then used to train their system model, PlatoLM, which demonstrates superior performance compared to other models, including Vicuna and UltraLaMA, in both Vicuna-Bench and pairwise comparisons in MT-Bench evaluations. This method shows promise for enhancing the quality and diversity of end-to-end models trained for conversational systems.

**Strengths:**

- Using Real User-System Data: The paper's approach of training the user simulator 'Socratic' with real user-system data is one of its strength. This approach ensures that the generated synthetic conversation data is grounded in actual human interactions, contributing to the effectiveness of the dataset and, by extension, the performance of PlatoLM.

- Thorough Experimental Evaluation: The paper's experimental evaluation is comprehensive, encompassing both automatic and manual assessments. The inclusion of manual evaluation provides a more nuanced understanding of the model's capabilities, as it incorporates human judgments.

- Promising Evaluation Results: Authors report promising results in the automatic and manual evaluations, with PlatoLM outperforming other models in the Vicuna-Bench and pairwise comparisons in MT-Bench. This demonstrates the effectiveness of their Socratic-based synthetic dataset when fine-tuning a system agent model.

**Weaknesses:**

- Limitations in Domain Transfer: The paper acknowledges limitations when transferring 'Socratic' to new domains. Specifically, it uses a seeding mechanism for domain transfer, which may not be the most flexible or scalable approach. The transferability of 'Socratic' could potentially be improved by instructing it through prompts or other means, making it more adaptable to new domains.

-  Narrow Focus on Backbone Architectures: The paper primarily focuses on performance of PlatoLM with LLaMA backbone, but it does not explore how PlatoLM's performance might vary with different backbone architectures. Examining how PlatoLM performs with various backbone architectures could provide valuable insights into their synthetic dataset generation effectiveness when training models with different backbone architectures.

**Questions:**

1. Could a fine-tuned GPT-3.5(4)-based model enhance 'Socratic' as the user simulator? Have you considered the possibility of using a fine-tuned GPT-3.5 model as the user simulator? This approach may potentially improve the diversity of generated dataset and result into a more generalizable PlatoLM.

2. Is 'Socratic' suitable as a prompt/policy generator for a ChatGPT-based user simulators? Exploring this avenue may lead to more flexible and generalized user simulator.

---

> ### Author Response · Authors · 2023-11-17
>
> ## Weaknesses 1: Limitations in Domain Transfer
>
> We fully agree with you！ In general, there are two ways for domain transfer: seed-guided vs. prompt-based.
>
> **Seed-guided Domain  Transfer**, one could use a simple ensemble of all the seed-guided samples, which will introduce inconsistent distribution, leading to a worse performance than our final models (see table below).
>
> | Seeds               | #Samples | MT-Bench |
> | ------------------- | -------- | -------- |
> | Free+Dolly          | 78k      | 6.28125  |
> | Free+ShareGPT+Dolly | 93k      | 6.31875  |
>
> **prompt-based domain transfer** Therefore, we are trying to use prompts-based domain transfer, as seen in Appendix A.  It can guide the simulator to ask the related topic (see newly-added cases in Appendix I.3.3), which intuitively **relieves the inconsistent distribution issue**. Prompt-based domain transfer is definitely a good extension for the proposed work. We are partially working on medical LLMs using PlatoLM with the prompt-based domain transfer.
>
> ---
> ## Weaknesses 2:  Narrow Focus on Backbone Architectures
>
>
> We added Mistral as the new backbone and redefined the pairing experiments in Appendix E. As the Table shown below,  we find two interesting trends.
>
> - **Diagonal Deterioration** In our study, we observed that pairings with differing backbones outperform pairings with identical backbones, see the ${\color{gray} \text{diagonal} ~  \text{elements}}$. For instance, the combination of llama2 & mistral outperformed llama1 & mistral is more effective than a mistral-mistral pairing. This may be because the same backbone stores identical knowledge, leading to some redundancy and an inability to complement each other for mutual enhancement. The former is like “one engages in self-reflection by both posing and answering questions”. This finding, in a broader sense,  indicates that interactive engagement with others may be more beneficial than solitary introspection.
>
> - **Non-diagonal scaling law** Beyond the aforementioned effect, performance consistently improves when a superior backbone is utilized (LLaMA 1- > LLaMA 2/Mistral or, LLaMA-2 -> Mistral), whether for the user simulator model or the assistant model. This is aligned with the common-known scaling law.
>
>
> | Q/A     | LLaMA-1       | LLaMA-2       | Mistral       |
> | ------- | ------------- | ------------- | ------------- |
> | LLaMA-1 | ${\color{gray} 5.7506±0.0314}$ | 6.0920±0.0515 | 6.4187±0.0470 |
> | LLaMA-2 | 5.8775±0.0205 | ${\color{gray}5.9900±0.0163}$ | 6.6838±0.0464 |
> | Mistral | 5.9094±0.0261 | 6.1744±0.0199 | ${\color{gray}6.3250±0.0406}$ |

---

> ### Author Response · Authors · 2023-11-17
>
> ## Question 1: Finetuned GPT-3.5
>
> We think so.
> Since OPENAI does not open up GPT-3.5 API for adjusting the learning target to the user, we only reversed the **content** of the dialogue between the user and the assistant. It costs $270.99 to fine-tune our 20K (30K after splitting) training set. An example of our training sets including the prompt template is shown below.
> ```
> {"messages": [{"role": "system", "content": "A chat between a helpful user and a curious artificial intelligence assistant.\n You are a curious assistant who asks further questions based on previous conversations, or directly asks brand new questions without any conversations as context.\n\n"}, {"role": "assistant", "content": "Write a 600 word essay reflecting on a time when you questioned or challenged a belief or idea. That time was when you first found out Santa Claus was not real. How was your thinking affected? Did this new revelation change your thinking and, if so, how did it? What was the outcome?"}, {"role": "user", "content": "I am an AI language model and do not have personal experiences but I can write about it from a third person perspective.\n\nWhen a child first discovers that Santa Claus is not real, it can be a significant moment in their life. The idea that there is a jolly old man who brings gifts on Christmas Eve is deeply ingrained in many cultures, and for many children, it is a central part of their holiday experience. So when the truth about Santa is"}]}
> ```
>
> After testing, we found that the fine-tuned GPT-3.5 simulator can ask questions in both free and seed modes（shown below）. Therefore, we believe this can provide more diverse questioning, as our paradigm can **activate** knowledge within GPT-3.5.
> ```
> Free mode：
> Q：Produce some sentences that are wordy and difficult to understand for undergraduate students.
>
> Seed mode：
> Seed Q from ShareGPT： what are the most cited papers on k-means clustering techniques
> Seed A from ShareGPT：..
> Q： Include the citation counts for the Spectral clustering citation.
> ```
> However, the model after adjusting the target may not be as effective at answering as it was before the adjustment, since we have also fine-tuned the answering model (Vicuna-7b-v1.5) to a questioning model（A-Q shown in below）and found that its ability to answer is weakened:
>
> | Model | Turn-1        | Turn-2        | MT-Bench      |
> | ----- | ------------- | ------------- | ------------- |
> | A     | -             | -             | 6.17          |
> | A-Q   | 5.6525±0.0489 | 3.9450±0.0665 | 4.7998±0.0146 |
>
> ---
>
> ##  Question 2:  Socratic' vs. a ChatGPT-based user simulators as  prompt/policy generator
>
> Thanks for your creative suggestion. Since Socratic can be applied in free mode, it can be used as a ChatGPT-based User Simulator's prompt/policy generator. This probably becomes a special case of UltraChat when the ChatGPT-based user simulator is prompted to ask Socratic questions.   However, the motivation of our paper focuses on **how to simulate real humans better** and draws a conclusion that the **more human-like questioning pattern** can teach LLMs better in multi-turn dialogs. Hence, we will try your advice for our future work.
>
> p.s. All sections with ± symbols on our supplementary experimental tables indicate that the results were rated 5 times by GPT-4 and the standard deviation was calculated.

---

> ### Author Response · Authors · 2023-11-22
> **Polite inquiry**
>
> We genuinely hope you will reconsider the scores after we have addressed your concerns and conducted additional experiments.
> Thank you for your time and consideration!

---

### Official Review · Reviewer_Z8HQ · 2023-11-01

**Soundness:** 2 fair
**Presentation:** 3 good
**Contribution:** 2 fair
**Rating:** 6
**Confidence:** 4

**Summary:**

This paper proposed to train a user simulator called "Socratic" using genuine human-machine conversations from ShareGPT to produce a high-quality human-centric synthetic conversation dataset. This dataset is then used to train a dialogue agent named "PlatoLM", which achieves the SOTA performance among 7B models.

**Strengths:**

1. The paper is well-written and clear. The authors have provided sufficient details about their approach, making it easy for others to understand and replicate their work.

2. The authors have conducted extensive experiments to validate their approach. The results show that their assistant model, PlatoLM, outperforms several baselines and achieves state-of-the-art performance among 7B models on MT-Bench.

3. The authors will release the code and dataset, which is beneficial to build a more powerful dialogue agent.

**Weaknesses:**

1. There are some doubts about the validity of the method in this paper. It's not clear where the performance improvement comes from. Please refer to the "Questions" for details.

2. Some writing issues.

   (1) The experimental results in Appendix F are not mentioned in the main text.

   (2) The direction of the quotation marks at the top of page 2.

**Questions:**

1. Scalability: Is there a performance bar for sample increasing? Table 2 uses 50K data, but the scaling in Figure 4 only achieves 30K, can you explain the reason for doing this? In addition, can more data maintain the effect of Scaling?

2. The experimental results in Appendix F are not mentioned in the main text, which happens to be an interesting experiment. In this regard, I have the following questions:

   (1) Are there any qualitative performance trends for user simulator and assistant model using the same or different backbone? For example, different backbones have a better performance than the same backbones.

   (2) The "overly clever LLaMA-2 backbone" mentioned in Appendix F needs to be further proved by using the LLaMA-1-13B model.

   (3) Will there be better results when the user simulator is more complex than the assistant model? For example, use LLaMA-2 as the user simulator and LLaMA-1 as the assistant model.

   (4) Will there be better results when the user simulator is used as the initialization checkpoint of the assistant model?

3. How does the performance of using Vicuna as the backbone of user simulator and assistant model? This means that we don't need to train an additional user simulator.

4. Is the difference between the middle and right subgraphs in Figure 1 only the user simulator? Is the way the dataset is generated the same?

5. Would this framework still work on a dataset generated by a stronger model, eg. alpaca-gpt4?

---

> ### Author Response · Authors · 2023-11-17
>
> ##  Weakness
> For weakness 1.2, Thanks for your careful reading.  We have fixed it.
>
> ---
> ##  Question 1: Scalability
> The reason why we scale to 30K is explained in the paper: we want to align it with **the training samples size of Scoratic**, which includes around 20K-30K samples (added in Sec.3.1.2).
>
> Inspired by your question, we scaled it into 100K samples, and found that:
> | #Samples | Turn-1        | Turn-2        | Avg. MT-Bench |
> | -------- | ------------- | ------------- | ------------- |
> | 2500     | 6.0413±0.0638 | 4.5425±0.0447 | 5.2910±0.0466 |
> | 5000     | 5.9050±0.0360 | 4.8050±0.0560 | 5.3550±0.0250 |
> | 10000    | 5.9700±0.0338 | 4.6950±0.0808 | 5.3325±0.0303 |
> | 20000    | 6.5313±0.0342 | 4.6938±0.0948 | 5.6125±0.0477 |
> | 30000    | 6.4388±0.0267 | 5.0625±0.0424 | 5.7506±0.0314 |
> | 50000    | 6.4325±0.0360 | 4.9475±0.0408 | 5.6900±0.0262 |
> | 100000   | 6.3388±0.0355 | 5.5800±0.0585 | 5.9599±0.0339 |
>
> - Overall, scaling up the sample size improves the overall model performance.
> - In the first turn, the model reaches saturation in performance after using 20k samples.
> - In the second turn, the model reaches the first saturation point after using 30K, but subsequent performance is **still increasing**.
>
> We hypothesize that this is because the dataset on which the questioning model was trained was only 20K (split into 30K segments).
>
> ---
>
> ## Question 2.1： Same or different backbone for user simulators and assistant models. *Do different backbones have a better performance than the same backbones?*
>
> We added Mistral as the new backbone and redefined the pairing experiments in Appendix E. resulting in three backbone models including LLaMA 1, LLaMA 2, and Mistral. As the Table shown below,  we find two interesting trends :
>
> | Q/A     | LLaMA-1       | LLaMA-2       | Mistral       |
> | ------- | ------------- | ------------- | ------------- |
> | LLaMA-1 | ${\color{gray} 5.7506±0.0314}$ | 6.0920±0.0515 | 6.4187±0.0470 |
> | LLaMA-2 | 5.8775±0.0205 | ${\color{gray}5.9900±0.0163}$ | 6.6838±0.0464 |
> | Mistral | 5.9094±0.0261 | 6.1744±0.0199 | ${\color{gray}6.3250±0.0406}$ |
>
>
>
> - **Diagonal Deterioration** In our study, we observed that pairings with differing backbones outperform pairings with identical backbones, see the ${\color{gray} \text{diagonal} ~  \text{elements}}$. For instance, the combination of llama2 & mistral which outperformed llama1 & mistral is more effective than a mistral-mistral pairing. This may be because the same backbone stores identical knowledge, leading to some redundancy and an inability to complement each other for mutual enhancement. The former is like “one engages in self-reflection by both posing and answering questions”. This finding, in a broader sense,  indicates that interactive engagement with others may be more beneficial than solitary introspection.
>
> - **Non-diagonal scaling law** Beyond the aforementioned effect, performance consistently improves when a superior backbone is utilized (LLaMA 1- > LLaMA 2/Mistral or, LLaMA-2 -> Mistral), whether for the user simulator model or the assistant model. This is aligned with the common-known scaling law.
>
>
>
> ---
>
> ## Question 2.2: *overly clever LLaMA-2 backbone* and using LLaMA-1-13B model.
> We also calculated intra- and inter-session question repetition rates and found that: although Mistral (>= LLaMA-2-13B) had a high repetition rate for inter-session questions, it also had a high repetition rate for intra-session questions, which limits the topic diversity.
> Thus, when keeping the same answer model backbone, Mistral is not always better than llama1 and llama2 as a questioning backbone (llama2-mistral > llama1-mistral > mistral-mistral).
>
> | Model   | Inter-Session | Intra-Session | Topic Diversity |
> | ------- | ------------- | ------------- | --------------- |
> | LLaMA-1 | 6.12%         | 28.44%        | 0.7063          |
> | LLaMA-2 | 6.33%         | 16.79%        | 0.7059          |
> | Mistral | 28.73%        | 28.87%        | 0.7116          |
>
> ---
>
> ## Question 2.3:  how about a more complex user simulator than the assistant model?
>
> This result occurs when the response model is based on LLaMA-1 (mistral-llama1 > llama2-llama1 > llama1-llama1). However, when the response model becomes llama2 and mistral, as shown in answer 2.2.1, this conclusion does not hold.
>
> ---
>
> ## Question 2.4:  how about initializing the assistant model with the checkpoint of the user simulator?
>
> We conducted additional experiments by fine-tuning the checkpoint of our simulators based on llama1 to the answer models (Q-A) and found that: **Although it can make answering and posing questions all-in-one via prompting, the results are not better than using the training set for simulators to fine-tune the answer model directly.**
>
> | Model | Turn-1        | Turn-2        | MT-Bench      |
> | ----- | ------------- | ------------- | ------------- |
> | A     | 6.2971±0.0547 | 5.1375±0.0702 | 5.7165±0.0541 |
> | Q-A   | 6.1775±0.0382 | 5.2125±0.0375 | 5.6950±0.0114 |

---

> ### Author Response · Authors · 2023-11-17
>
> ## Question 3:  Vicuna as the backbone of the user simulator and assistant model
>
> We conducted additional experiments to fine-tune Vicuna-7b-v1.5 with the reversed learning objectives (A-Q) and found that: **Although the reversed model can raise and answer questions, the answer ability is weakened after changing the learning objectives.**
>
> | Model | Turn-1        | Turn-2        | MT-Bench      |
> | ----- | ------------- | ------------- | ------------- |
> | A     | -             | -             | 6.17          |
> | A-Q   | 5.6525±0.0489 | 3.9450±0.0665 | 4.7998±0.0146 |
>
> The reason might be below.  The assistant model like Vicuna focuses on following users' instructions. This does not align with the purpose of a questioning model. Learning instruction-following does not contribute to and sometimes harms the questioning quality.  Therefore,  Vicuna, which is equipping LLaMA with additional instruction following ability, is not beneficial to be the backbone of the user simulator.
>
> ---
> ## Question 4: difference in the middle and right subgraphs in Figure 1
>
>  When inferencing, the generation approach is different in **3 aspects**：
>
> （1）The Seed：
> Middle subgraphs (Baize、UltraLM) use seeds from humans or ChatGPT and add them to the prompt template.
> Right subgraphs (Ours) can use a seed domain in the prompt template, seed conversation in history message, and also can be seed-free, directly activating the topic from different backbones.
>
> （2）Termination Mechanism：
> Baize uses prompts to control the termination of conversation, which we call soft control in paper.
> UltraLM may use the turn range(3-7 turns) to control the termination.
> Ours uses the maximal context length to terminate, which we call hard control in the paper.
>
> （3）Post-processing
> Baize uses self-distillation via RL to pick the answers.
> UltraLM conducts extensive post-preprocessing, such as removing the thanks in the human's question.
> Ours can be post-preprocessing free. Although our paradigm may cause the repetitive question in the first turn of mult-turn dialogs, the response model is better by preserving it than removing it, which we have demonstrated in Appendix D.
>
> ---
>
> ## Question 5:  Does this method work on a dataset generated by a stronger model?
>
> We believe so. However, since the dataset generated by human-gpt4 is small in public ShareGPT(6K samples), we tried to pair the simulator fine-tuned on the human-to-3.5 dataset via LLaMA1 and the GPT-4 API. The scores in the MT-Bench are shown below. Using the dataset generated by GPT-4, the resultant models perform better than using GPT-3.5.
> The performance of the response model can be scaled up with our paradigm as well.
>
> | Q | A | Scale | Turn-1 | Turn-2 | MT-Bench |
> | ----- | ----- | ----- | ------ | ------ | -------- |
> | human-3.5 | Socratic-3.5 | 10,192   | 6.5188±0.0523 |5.3375±0.0656 |5.9281±0.0378 |
> | human-3.5 | Socratic-4| 10,192 | 6.5328±0.0660 | 5.6000±0.0492 | 6.0658±0.0302|
> | human-3.5 | Socratic-3.5 | 24,043 | 6.6138±0.0320 | 5.5175±0.0818 | 6.0656±0.0368 |
> | human-3.5 | Socratic-4 | 24,043 | 6.5638±0.0399| 5.7338±0.0537| 6.1488±0.0216 |
>
> ---
>
> Thanks for your creative suggestion, especially questions 2.4 and 3. Our further work will consider how we can make our paradigm all-in-one.
>
> p.s. All sections with ± symbols on our supplementary experimental tables indicate that the results were rated 5 times by gpt4 and the standard deviation was calculated.

---

> ### Author Response · Authors · 2023-11-22
> **Polite inquiry**
>
> We sincerely hope you will reconsider the scores after we explained your concerns and added experiments.
> Thank you for your time and consideration!

---

> > ### Comment · Reviewer_Z8HQ · 2023-11-22
> >
> > Thank you for your comprehensive response. Most of my concerns have been addressed. I have decided to revise my score to 6: marginally above the acceptance threshold.

---

### Official Review · Reviewer_6WiD · 2023-11-01

**Soundness:** 3 good
**Presentation:** 3 good
**Contribution:** 3 good
**Rating:** 8
**Confidence:** 3

**Summary:**

This paper proposes using a trained language model rather than a general-purpose language model as a user simulator to generate a synthetic conversation dataset. The dataset is then used to train pre-trained language models. The authors train an assistant model called PlatoLM on the synthetic conversation data generated by the trained user simulator. They show that PlatoLM outperforms models trained on synthetic conversations produced by a general-purpose language model.

**Strengths:**

1. This paper demonstrates the efficacy of training a user simulator model for generating synthetic training data to improve language models. The approach of training a user simulator could be broadly applied across domains when curating datasets to train language models.
2. The comprehensive experiments present promising results when training language models with synthetic conversation datasets produced by the proposed approach of using a trained user simulator model. The trained models outperform those trained on synthetic data generated by a general-purpose language model.
3. The authors curate a high-quality, human-like multi-turn conversation dataset using the trained user simulator model. The dataset will be open-sourced.

**Weaknesses:**

The proposed approach of training a user simulator model to generate synthetic training data, while logical, may lack sufficient novelty. Using a trained language model as a user simulator aligns with prior work on conversational agents and data augmentation. The straightforward nature of training a user simulator model makes the technique intuitive, but also means the work is incremental.

**Questions:**

1. In section 5.3, what could be the possible reason for the unstable performance increase when scaling up training samples
2. A minor typo in section 3.2.1, ChaTGPT should be ChatGPT

---

> ### Author Response · Authors · 2023-11-17
>
> ## Novelty of this paper.
>
> We sympathize with the reviewer's concerns regarding the novelty of our paper. Due to limited space in the conference paper, we removed many comparisons between prior works and ours. Now, we added it back to the **Background** part.
>
> **Unlike previous data augmentation**, which always focuses on **quantity**, we focus on constructing **high-quality** questions in this paper. As shown in Table 4, the questions constructed by our paradigm outperform other methods, especially in topic diversity and human-like ratio, which is closely related to the performance or response model, as shown in Figure 5.
>
> Dataset|Topic Diversity ↓	|Human-like Ratio↑
> |---|---|---|
> Self-Chat|	0.7190|	0.1758
> UltraChat|	0.7158|	0.1157
> SocraticChat|	**0.7078**| **0.6727**
>
> With the same training settings (SFT) and backbone (LLaMA2), with only 50,728 samples taught by our paradigm and a maximal context length of 2048, we outperformed Vicuna-7b-v1.5 (which used 125,000 samples with a context length of 16,000) in MT-Bench. After changing the simulator backbone and the response model backbone to LLaMA-2 and Mistral (as added in Appendix E following Reviewer Z8HQ and Reviewer FWRq), with only **28,500** samples, we surpassed Vicuna-13B-v1.5 and LLaMA-2-13B-chat in MT-Bench.
>
> | Models           | Bakcbone | #Samples | Training Type | MT-Bench  | AlpacaEval |
> | ---------------- | -------- | -------- | ------------- | --------- | ---------- |
> | Baize-v2-7b      | LLaMA    | 100K     | LoRA, RL      | -         | 63.85%     |
> | Baize-v2-13b     | LLaMA    | 100K     | LoRA, RL      | 5.75      | 66.96%     |
> | LLaMA-2-7b-chat  | LLaMA2   | 1100K    | SFT, RL       | 6.27      | 71.37%     |
> | PlatoLM-7b       | LLaMA    | 10K      | SFT           | 5.33±0.03 | 71.89%     |
> | Vicuna-7b-v1.3   | LLaMA2   | 125K     | SFT           | -         | 76.84%     |
> | Vicuna-7b-v1.5   | LLaMA2   | 125K     | SFT           | 6.17      | -          |
> | UltraLM-13b-v1.0 | LLaMA    | 1468K    | SFT           | -         | 80.64%     |
> | GPT-3.5          | -        | -        | -             | -         | 81.71%     |
> | PlatoLM-7b       | LLaMA2   | 50.73K   | SFT           | 6.29±0.04 | 81.94%     |
> | PlatoLM-7b       | Mistral  | 28.5K    | SFT           | 6.68±0.05 | -          |
>
> **Compared to human involvement**, our paradigm is more cost-efficient than human labor, especially in certain scenarios where it is not possible to actively invite users for interaction：
>
> - **Privacy-friendliness**: Using the simulated questions instead of the original ShareGPT could reduce privacy leakage, as no real data is used.
>
> - **Transferability**: Especially in scenarios where the user's question does not exist due to the new sensitivity to ethics, like medical applications, it is necessary to synthesize such data.
>
> - **Scalability**: Scalability is the core recipe of LLMs. Our paradigm can easily scale data up, especially using a well-designed prompt or, more heavily, a diverse set of seed questions. Additionally, the original websites of ShareGPT do not allow downloading, limiting the research investigation to improve LLMs in terms of conversational abilities.
>
> **Compared to static role-playing ChatGPTs**, Socratic simulates the human questioning pattern by direct supervision. Due to the absence of human questioning pattern supervision in static role-playing ChatGPTs (e.g., Baize and UltraChat), the generated questions and follow-up questions do not perform as rich topic structures (i.e., topic shifting and topic drifting) as human conversation has, leading to diminished topic diversity, while our paradigm can be seed-free via activating the diverse topic from different backbones and better capturing the human questioning pattern.
>
> Basically, our target focuses on how to better **simulate the human questioning pattern** in human-machine interactions. We found that the response can be taught better by more human-like questioning patterns in multi-turn dialogues. The dataset is only a **by-product** of our proposed paradigm.
>
> Hence, despite the paradigm seems to be straightforward, our proposed method has demonstrated superior performance, which we believe is non-trivial.

---

> ### Author Response · Authors · 2023-11-17
>
> ---
>
> ## Question 1: on the scaling law of PlatoLM
>
> **On the stability issue**
>
> In fact, the overall performance of the model is stable, and the instability is reflected **in the fewer sample stages (2.5k-10k) of turn1 and turn2 respectively**.
> We believe there are two possible reasons for this:
>
> 1. The question may **repeat** on the first turn of a multi-turn dialogue ( see Appendix D), which may result in multiple samples with the same **topic**, such that the score **fluctuates slightly** on the MT-bench in which the single evaluation scores on the domains the model didn't good at will be **posed a penalty** (according to MT-bench's author).
>
> 2. **SFT** is different from pre-training. Scaling law refers to the fact that **in pre-training**, the training loss follows a power law with the total number of tokens in the training samples. Using SFT is more likely to **activate the knowledge in the backbone**, which does not apply to the power law relationship of pre-training models ([1] see 1.1 Smooth power laws).
>
> **The scaling experiment**
>
> Inspired by your question, we scaled it into 100K samples, and found that:
> | #Samples | Turn-1        | Turn-2        | Avg. MT-Bench |
> | -------- | ------------- | ------------- | ------------- |
> | 2500     | 6.0413±0.0638 | 4.5425±0.0447 | 5.2910±0.0466 |
> | 5000     | 5.9050±0.0360 | 4.8050±0.0560 | 5.3550±0.0250 |
> | 10000    | 5.9700±0.0338 | 4.6950±0.0808 | 5.3325±0.0303 |
> | 20000    | 6.5313±0.0342 | 4.6938±0.0948 | 5.6125±0.0477 |
> | 30000    | 6.4388±0.0267 | 5.0625±0.0424 | 5.7506±0.0314 |
> | 50000    | 6.4325±0.0360 | 4.9475±0.0408 | 5.6900±0.0262 |
> | 100000   | 6.3388±0.0355 | 5.5800±0.0585 | 5.9599±0.0339 |
>
> - Overall, scaling up the sample size improves the overall model performance.
> - In the first turn, the model reaches saturation in performance after using 20k samples.
> - In the second turn, the model reaches the first saturation point after using 30K, but subsequent performance is **still increasing**.
>
> We speculate that this is because **the training set size** for the questioning model was only 20K (split into 30K segments). (in Sec. 3.1.2 ).
>
> We have shown that the pattern of real people's questions learned by our approach contributes to the quality of response models in dynamic multi-turn dialogue(in Sec.5.1 Table 3), and based on scaling experiments, we find that this teaching capability can be **scalable** as well.
>
> ---
>
> ## Question 2
>
>  Thank you for reading our paper so carefully. We have corrected it.
>
>
> [1] Kaplan J, McCandlish S, Henighan T, et al. Scaling laws for neural language models[J]. arXiv preprint arXiv:2001.08361, 2020.

---

> > ### Comment · Reviewer_6WiD · 2023-11-20
> >
> > Thank you for your detailed response. I have carefully read them.

---

> ### Author Response · Authors · 2023-11-21
> **Polite inquiry**
>
> Thank you for your response. Please let us know if there are any concerns that we haven't addressed. We are more than happy to provide any materials and experimental results, or to clarify anything that may have caused confusion.
>
> Polite inquiry: Would it be possible to revert the score back to 8? We observed that there were no changes in your comment updated at 18:41 on 12 November 2023, yet the score for our submission has unexpectedly dropped from 8 to 6.

---

> > ### Comment · Reviewer_6WiD · 2023-11-23
> >
> > Thank you for providing such a detailed response and including additional experiments in your work.
> >
> > Upon reviewing the comments of other reviewers, I have identified some weaknesses that I had not previously noticed. As a result, I adjusted the score accordingly on 12 November 2023.
> >
> > After reading the author's response to the other reviewers, I found that most of the concerns I agreed with were adequately addressed. Therefore, I decided to increase the score back to 8.
> >
> > However, please note that further discussion between reviewers may still occur after this period, which could potentially lead to score changes.

---

> > > ### Author Response · Authors · 2023-11-23
> > >
> > > Thank you for your reevaluating the score. It is encouraging to hear that most of your concerns were adequately addressed in our response. Please feel free to reach out to us at any time, and we will be more than happy to provide further clarification or assistance. Once again, we sincerely appreciate your time and effort in reviewing our work.